# A flexible model for training action localization with varying levels of supervision

**Guilhem Chéron**[* 1 2]    **Jean-Baptiste Alayrac**[* 1]    **Ivan Laptev**[1]    **Cordelia Schmid**[2]

## Abstract

Spatio-temporal action detection in videos is typically addressed in a fully-supervised setup with manual annotation of training videos required at every frame. Since such annotation is extremely tedious and prohibits scalability, there is a clear need to minimize the amount of manual supervision. In this work we propose a unifying framework that can handle and combine varying types of less-demanding *weak supervision*. Our model is based on discriminative clustering and integrates different types of supervision as constraints on the optimization. We investigate applications of such a model to training setups with alternative supervisory signals ranging from video-level class labels to the full per-frame annotation of action bounding boxes. Experiments on the challenging UCF101-24 and DALY datasets demonstrate competitive performance of our method at a fraction of supervision used by previous methods. The flexibility of our model enables joint learning from data with different levels of annotation. Experimental results demonstrate a significant gain by adding a few fully supervised examples to otherwise weakly labeled videos.

## 1   Introduction

Action localization aims to find spatial and temporal extents as well as classes of actions in the video, answering questions such as *what* are the performed actions? *when* do they happen? and *where* do they take place? This is a challenging task with many potential applications in surveillance, autonomous driving, video description and search. To address this challenge, a number of successful methods have been proposed [11, 13, 18, 31, 36, 37, 45]. Such methods, however, typically rely on exhaustive supervision where each frame of a training action has to be manually annotated with an action bounding box.

Manual annotation of video frames is extremely tedious. Moreover, achieving consensus when annotating action intervals is often problematic due to ambiguities in the start and end times of an action. This prevents fully-supervised methods from scaling to many action classes and training from many examples. To avoid exhaustive annotation, recent works have developed several weakly-supervised methods. For example, [46] learns action localization from a sparse set of frames with annotated bounding boxes. [27] reduces bounding box annotation to a single spatial point specified for each frame of an action. Such methods, however, are designed for particular types of weak supervision and can be directly used neither to compare nor to combine various types of annotation.

In this work we design a unifying framework for handling various levels of supervision[†]. Our model is based on discriminative clustering and integrates different types of supervision in a form of

[*]Equal contribution.
[†]Project webpage `https://www.di.ens.fr/willow/research/weakactionloc/`.

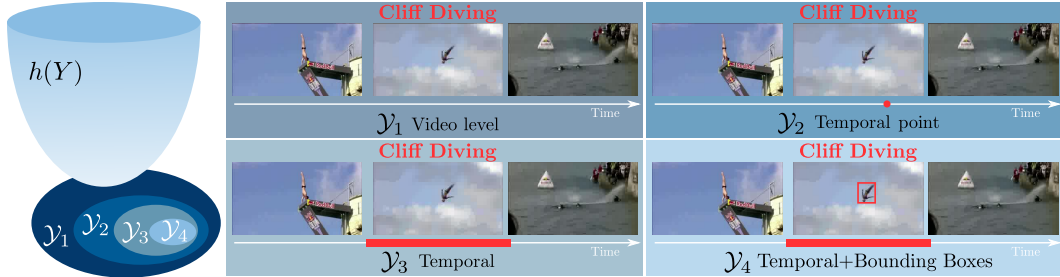

Figure 1: **A flexible method for handling varying levels of supervision.** Our method estimates a matrix $Y$ assigning human tracklets to action labels in training videos by optimizing an objective function $h(Y)$ under constraints $\mathcal{Y}_s$. Different types of supervision define particular constraints $\mathcal{Y}_s$ and do not affect the form of the objective function. The increasing level of supervision imposes stricter constraints, e.g. $\mathcal{Y}_1 \supset \mathcal{Y}_2 \supset \mathcal{Y}_3 \supset \mathcal{Y}_4$ as illustrated for the Cliff Diving example above.

optimization constraints as illustrated in Figure 1. We investigate applications of such a model to training setups with alternative supervisory signals ranging from video-level class labels over temporal points or sparse action bounding boxes to the full per-frame annotation of action bounding boxes. Experiments on the challenging UCF101-24 and DALY datasets demonstrate competitive performance of our method at a fraction of supervision used by previous methods. We also demonstrate a significant gain by adding a few fully supervised examples to weakly-labeled videos. In summary, the main contributions of our work are (i) a flexible model with ability to adopt and combine various types of supervision for action localization and (ii) an experimental study demonstrating the strengths and weaknesses of a wide range of supervisory signals and their combinations.

## 2 Related work

**Spatio-temporal action localization** consists in finding action instances in a video volume, i.e. both in space and time. Initial attempts [20, 22] scanned the video clip with a 3D sliding window detector on top of volumetric features. Next, [29, 43] adapted the idea of object proposals [2, 42] to video action proposals. Currently, the dominant strategy [11, 31, 36, 37, 45] is to obtain per-frame action detections and then to link them into continuous spatio-temporal tracks. The most recent methods [13, 15, 18, 35, 49] operate on multiple frames and leverage temporal information to determine actions. Examples include stacking features from several frames [18] and applying the I3D features [6] to spatio-temporal volumes [13]. These methods are fully supervised and necessitate a large quantity of annotations.

**Weakly supervised learning for action understanding** is promising since it can enable a significant reduction of required annotation efforts. Prior work has explored the use of readily-available sources of information such as movie scripts [4, 8] to discover actions in clips using text analysis. Recent work has also explored more complex forms of weak supervision such as the ordering of actions [5, 16, 33].

A number of approaches address temporal action detection with weak supervision. The Untrimmed-Net [44] and Hide-and-Seek [38] methods are the current state of the art. In [44], the authors introduce a feed forward neural network composed of two branches – one for classification and another one for selecting relevant frames – that can be trained end-to-end from clip level supervision only. The method proposed in [38] obtains more precise object boxes or temporal action boundaries by hiding random patches of the input images at training time, forcing the network to focus on less discriminative parts.

In contrast, we seek to not only localize the action temporally but to also look for spatial localization from weak supervision. In [40, 48], the authors propose unsupervised methods to localize actions by using graph clustering algorithms to first discover action classes and then localize actions within each cluster. Other works assume that clip level annotations are available [7, 39]. These methods often rely on action proposals, followed by a selection procedure to find the most relevant segment. Contrary to this line of work, we do not use action proposals. We rely instead on recent advances in off-the-shell human detectors [10] and use human tracks. More recently [26] makes use of pseudo annotation, to integrate biases (e.g. presence of objects, the fact that the action is often in the center of the video...) to further improve performance. Most of this work uses Multiple Instance Learning (MIL) in order to select discriminative instances from the set of all potential candidates. Here, we

instead use discriminative clustering, which does not require EM optimization but instead relies on a convex relaxation with convergence guarantees.

Most related to us, are the works from [27] and [46] studying the trade-off between the annotation cost and final performance. [27] shows that one can simply use spatial points instead of bounding boxes and still obtain reasonable performance. [46] instead demonstrates that only a few frames with bounding box annotation are necessary for good performance. Here, we introduce a method that can leverage varying levels of supervision.

**Discriminative clustering** [3, 47] is a learning method that consists in clustering the data so that the separation is easily recoverable by a classifier. In this paper, we employ the method proposed in [3] which is appealing due to its simple analytic form, its flexibility and its recent successes for weakly supervised methods in computer vision [1, 4, 17]. To use this method, we rely on a convex relaxation technique which transforms the initial NP-hard problem into a convex quadratic program under linear constraints. The Frank-Wolfe algorithm [9] has shown to be very effective for solving such problems. Recently, [28] proposed a block coordinate [21] version of Frank-Wolfe for discriminative clustering. We leverage this algorithm to scale our method to hundreds of videos.

## 3 Problem formulation and approach

We are given a set of $N$ videos of people performing various actions. The total number of possible actions is $K$. Our goal is to learn a model for each action, to localize actions in time and space in *unseen* videos. Given the difficulty of manual action annotation, we investigate how different types and amounts of supervision affect the performance of action localization. To this end, we propose a model that can adopt and combine various levels of spatio-temporal action supervision such as (i) video-level only annotations, (ii) a single temporal point, (iii) a single bounding box, (iv) temporal bounds only, (v) temporal bounds with few bounding boxes and (vi) full supervision. In all cases, we are given tracks of people, i.e. sequences of bounding box detections linked through time. These tracks are subdivided into short elementary segments in order to obtain precise action boundaries and to allow the same person to perform multiple actions. These segments are called *tracklets* and our goal is to assign them to action labels. In total, we have $M$ such tracklets.

**A single flexible model: one cost function, different constraints.** We introduce a single model that can be trained with various levels and amounts of supervision. The general idea is the following. We learn a model for action classification on the train set from *weak supervision* using discriminative clustering [3, 47]. This model is used to predict spatio-temporal localization on test videos.

The idea behind discriminative clustering is to cluster the data (e.g. assign human tracklets to action labels) so that the clusters can be recovered by a classifier given some feature representation. The clustering and the classifier are simultaneously optimized subject to constraints that can regularize the problem and guide the solution towards a preferred direction. In this work we use constraints as a mean to encode different types of supervisory signals. More formally, we frame our problem as recovering an assignment matrix of tracklets to actions $Y \in \{0,1\}^{M \times K}$ which minimizes a clustering cost $h$ under constraints:

$$\min_{Y \in \mathcal{Y}_s} h(Y), \tag{1}$$

where $\mathcal{Y}_s$ is a constraint set encoding the supervision available at training time. In this work we consider constraints corresponding to the following types of supervision.

(i) **Video level action labels**: only video-level action labels are known.

(ii) **Single temporal point**: we have a rough idea when the action occurs, but we do not have either the exact temporal extent or the spatial extent of the action. Here, we can for example build our constraints such that at least one human track (composed of several tracklets) should be assigned to the associated action class in the neighborhood of a temporal point.

(iii) **One bounding box (BB)**: we are given the spatial location of a person at a given time inside each action instance. Spatial constraints force tracks that overlap with the bounding box to match the action class around this time step.

(iv) **Temporal bounds**: we know *when* the action occurs but its spatial extent is unknown. We constrain the labels so that at least one human track contained in the given temporal

interval is assigned to that action. Samples outside annotated intervals are assigned to the background class.

(v) **Temporal bounds with bounding boxes (BBs)**: combination of (iii) and (iv).

(vi) **Fully supervised**: annotation is defined by the bounding box at each frame of an action.

All these constraints can be formulated under a common mathematical framework that we describe next. In this paper we limit ourselves to the case where each tracklet should be assigned to only one action class (or the background). More formally, this can be written as follows:

$$\forall\, m \in [1 \mathinner{\ldotp\ldotp} M], \quad \sum_{k=1}^{K} Y_{mk} = 1. \tag{2}$$

This does not prevent us from having multiple action classes for a video, or to assign multiple classes to tracklets from the same human track. Also note that it is possible to handle the case where a single tracklet can be assigned to multiple actions by simply replacing the equality with, e.g., a greater or equal inequality. However, as the datasets considered in this work always satisfy (2), we keep that assumption for the sake of simplicity.

We propose to enforce various levels of supervision with two types of constraints on the assignment matrix $Y$ as described below.

**Strong supervision with equality constraints.** In many cases, even when dealing with weak supervision, the supervision may provide strong cues about a tracklet. For example, if we know that a video corresponds to the action 'Diving', we can assume that no tracklet of that video corresponds to the action 'Tennis Swing'. This can be imposed by setting to $0$ the corresponding entry in the assignment matrix $Y$. Similarly, if we have a tracklet that is outside an annotated action interval, we know that this tracklet should belong to the background class. Such cues can be enforced by setting to $1$ the matching entry in $Y$. Formally,

$$\forall (t, k) \in \mathcal{O}_s \quad Y_{tk} = 1, \quad \text{and} \quad \forall (t, k) \in \mathcal{Z}_s \quad Y_{tk} = 0, \tag{3}$$

with $\mathcal{O}_s$ and $\mathcal{Z}_s$ containing all the tracklet/action pairs that we want to match ($\mathcal{O}_s$) or dissociate ($\mathcal{Z}_s$).

**Weak supervision with at-least-one constraints.** Often, we are uncertain about which tracklet should be assigned to a given action $k$. For example, we might know when the action occurs, without knowing where it happens. Hence, multiple tracklets might overlap with that action in time but not all are good candidates. These multiple tracklets compose a bag [4], that we denote $\mathcal{A}_k$. Among them, we want to find at least one tracklet that matches the action, which can be written as:

$$\sum_{t \in \mathcal{A}_k} Y_{tk} \geq 1. \tag{4}$$

We denote by $\mathcal{B}_s$ the set of all such bags. Hence, $\mathcal{Y}_s$ is characterized by defining its corresponding strong supervision $\mathcal{O}_s$ and $\mathcal{Z}_s$ and the bags $\mathcal{B}_s$ that compose its weak supervision.

**Discriminative clustering cost.** As stated above, the intuition behind discriminative clustering is to separate the data so that the clustering is easily recoverable by a classifier over the input features. Here, we use the square loss and a linear classifier, which corresponds to the DIFFRAC setting [3]:

$$h(Y) = \min_{W \in \mathbb{R}^{d \times K}} \frac{1}{2M} \|XW - Y\|_F^2 + \frac{\lambda}{2} \|W\|_F^2. \tag{5}$$

$X \in \mathbb{R}^{M \times d}$ contains the features describing each tracklets. $W \in \mathbb{R}^{d \times K}$ corresponds to the classifier weights for each action. $\lambda \in \mathbb{R}^+$ is a regularization parameter. $\|.\|_F$ is the standard Frobenius matrix norm. Following [3], we can solve the minimization in $W$ in closed form to obtain: $h(Y) = \frac{1}{2M}\mathrm{Tr}(YY^T B)$, where $\mathrm{Tr}(\cdot)$ is the matrix trace and $B$ is a strictly positive definite matrix (hence $h$ is strongly convex) defined as $B := I_M - X(X^T X + M\lambda I_d)^{-1}X^T$. $I_d$ stands for the $d$-dimensional identity matrix.

**Optimization.** Directly optimizing the problem defined in (1) is NP hard due to the integer constraints. To address this challenge, we follow recent approaches such as [5] and propose a convex relaxation of the constraints. The problem hence becomes $\min_{Y \in \bar{\mathcal{Y}}_s} h(Y)$, where $\bar{\mathcal{Y}}_s$ is the convex hull of $\mathcal{Y}_s$. To deal with such constraints we use the Frank-Wolfe algorithm [9], which has the nice property of only requiring to know how to solve linear programs over the constraint set. Another challenge lies in the fact that we deal with large datasets containing hundreds of long videos. Using the fact that our set of constraints actually decomposes over the videos, $\bar{\mathcal{Y}}_s = \bar{\mathcal{Y}}_s^1 \times \cdots \times \bar{\mathcal{Y}}_s^N$, we use the block coordinate Frank-Wolfe algorithm [21] that has been recently adapted to the discriminative clustering objective [28]. This allows us to scale to several hundreds of videos.

## 4 Experiments

### 4.1 Datasets and metrics

**The UCF101 dataset** [41] was originally designed for action classification. It contains 13321 videos of 101 different action classes. A subset of 24 action classes (selected by [12]) was defined for the particular task of spatio-temporal action localization in 3207 videos. We refer to this subset as 'UCF101-24'. The videos are relatively short and the actions usually last a large part of the duration (at least 80% of the video length for half of the classes). The annotation is exhaustive, i.e., for each action instance, the full person track within the temporal interval of an action is annotated. We use recently corrected ground truth tracks [36]. Each UCF101-24 video contains actions of a single class. There is only one split containing 2293 train videos and 914 test videos.

**The DALY dataset** [46] is a recent large-scale dataset for action localization, with 10 different daily actions (e.g. 'drinking', 'phoning', 'cleaning floor'). It contains 510 videos for a total of 31 hours arranged in a single split containing 31 train and 20 test videos per class. The average length of the videos is 3min 45s. They are composed of several shots, and all actions are short w.r.t. the full video length, making the task of temporal action localization very challenging. Also, DALY may contain multiple action classes in the same video. For each action instance, its temporal extent is provided while bounding boxes are spatially annotated for a sparse set of frames within action intervals.

**Performance evaluation.** To evaluate the detection performance, we use the standard spatio-temporal intersection over union (IoU) criterion between a candidate track and a ground-truth track. It is defined for the UCF101-24 benchmark as the product of the temporal IoU between the time segments of the tracks and the average spatial IoU on the frames where both tracks are present. The candidate detection is correct if its intersection with the ground-truth track is above a threshold (in our experiments, set to 0.2 or 0.5) and if both tracks belong to the same action class. Duplicate detections are considered as false positives. The overall performance is evaluated in terms of mean average precision (mAP). The *same* metric is used for all levels of supervision which enables a fair comparison. Note that since only sparse spatial annotation is available for the DALY dataset, its benchmark computes spatial IoU at the annotated frames location only while temporal IoU remains the same.

### 4.2 Implementation details

The code of the paper to reproduce our experiments can be found on the webpage of the project[*].

**Person detector and tracker.** Person boxes are obtained with Faster-RCNN [32] using the ResNet-101 architecture [14]. When no spatial annotation is provided, we use an off-the-shelf person detector trained on the COCO dataset [23]. Otherwise, action detections are obtained by training the detector on the available frames with bounding boxes starting from ImageNet [34] pre-training. We use the implementation from the Detectron package [10]. Detections from the human detector are linked into continuous tracks using KLT [24] to differentiate between person instances. Class-specific detections are temporally aggregated by a simpler online linker [18, 37] based on action scores and overlap.

**Feature representation.** In our model, person tracks are divided into consecutive subtracks of 8 frames which we call *tracklets*. Due to its recent success for action recognition, we use the I3D [6] network trained on the Kinetics dataset [19] to obtain tracklet features. More precisely, we extract video descriptors with I3D RGB and flow networks after the 7-th inception block, before max-pooling, to balance the depth and spatial resolution. The temporal receptive field is 63 frames and the temporal

---

[*]https://www.di.ens.fr/willow/research/weakactionloc/

stride is 4 frames. The input frames are resized to $320 \times 240$ pixels, which results in feature maps of size $20 \times 15$ with 832 channels. We use ROI-pooling to extract a 832-dimensional descriptor for each human box at a given time step. The final 1664-dimensional representation is obtained by concatenating the descriptor from the RGB and the flow streams. Finally, a tracklet representation is obtained by averaging the descriptors corresponding to the detections spanned by this tracklet.

**Optimization.** For all types of supervision we run the block-coordinate Frank-Wolfe optimizer for 30k iterations (one iteration deals with one video). We use optimal line search and we sample videos according to the block gap values [30] which speeds up convergence in practice.

**Temporal localization.** At test time, person tracks have to be trimmed to localize the action instances temporally. To do so, each of the $T$ detections composing a track are scored by the learnt classifier. Then, the scores are smoothed out with a median filtering (with a window of size 25), giving, for an action $k \in [1..K]$, the sequence of detection scores $s^k = (s_1^k, ..., s_T^k)$. Within this track, temporal segmentation is performed by selecting consecutive person boxes with scores $s_t^k > \theta_k$, leading to subtrack candidates for action $k$. A single person track can therefore produce several spatio-temporal predictions at different time steps. A score for each subtrack is obtained by averaging corresponding detection scores. Finally, non-maximum-suppression (NMS) is applied to eliminate multiple overlapping predictions with spatio-temporal IoU above 0.2.

**Hyperparameters.** The regularization parameter $\lambda$ (see equation (5)) is set to $10^{-4}$ in all of our experiments. To calibrate the model, the temporal localization thresholds $\theta_k$ are validated per class on a separate set corresponding to $10\%$ of the training set.

**Computational cost.** At each training iteration of our algorithm, the computational complexity is linear in the number of tracklets for a given video. When running 30K iterations (sufficient to get stable performance in terms of training loss) on a relatively modern computer (12 cores and 50Go of RAM) using a single CPU thread, our python implementation takes 20 minutes for UCF101-24 (100K tracklets) and 60 minutes for DALY (500K tracklets) for the 'Temporal + 1 BB' setting. Note that the timings are similar across different levels of supervision.

### 4.3 Supervision as constraints

In this section, we describe in details the constraint sets $\mathcal{Y}_s$ for different levels of supervision considered in our work. Recall from Section 3 that $\mathcal{Y}_s$ is characterized by $\mathcal{O}_s$ and $\mathcal{Z}_s$, the sets of tracklets for which we have strong supervision, and by $\mathcal{B}_s$, the set of bags containing tracklets that are candidates to match a given action. Note that in all settings we have one class that corresponds to the background class. In the following, a *time unit* corresponds to one of the bins obtained after uniformly dividing a time interval. Its size is 8 frames.

**Video level.** Here, $\mathcal{B}_s$ contains as many bags as action *classes* occurring in the entire video. Every bag contains all the tracklets of the video. This constrains each annotated action class to be assigned to at least one tracklet from the video. We also construct $\mathcal{Z}_s$ making sure no tracklets are assigned to action classes not present in the video.

**Shot level.** This setting is identical to the '**Video level**' one, but we further decompose the video into clips and assume we have clip-level annotations. Such clips are obtained by the shot detection [25] for the DALY dataset. Since UCF101-24 videos are already relatively short, we do not report Shot level results for this dataset.

**Temporal point.** For each action instance, we are given a point in time which falls into the temporal interval of the action. In our experiments, we sample this time point uniformly within the ground truth interval. This corresponds to the scenario where an annotator would simply click one point in time where the action occurs, instead of precisely specifying the time boundaries. We then create a candidate interval around that time point with a fixed size of 50 frames (2 seconds). This interval is discretized into time units. For each of them, we impose that at least one tracklet should be assigned to the action instance. Hence, we add $A \times U$ bags to $\mathcal{B}_s$, where $A$ is the number of instances and $U$ is the number of time units in a 50 frames interval. Finally, $\mathcal{Z}_s$ is built as before: no tracklet should match actions absent from the video.

**One bounding box.** At each location of the previous temporal points, we are now additionally given the corresponding action instance bounding box (a.k.a. keyframe in [46]). Similarly, we construct 50 frames time intervals. We consider all tracklets whose original track has a temporal overlap with the

annotated frame. Then, we compute the spatial IoU between the bounding box of the track at the time of the annotated frame and the bounding box annotation. If this IoU is less than 0.3 for all possible frames, we then construct $\mathcal{O}_s$ so that these tracklets are forced to the background class. Otherwise, if the tracklet is inside the 50 frames time interval, we force it to belong to the action class instance with the highest IoU by augmenting $\mathcal{O}_s$. Again, for each time unit of the interval, we construct $\mathcal{B}_s$ such that at least one tracklet matches the action instance. $\mathcal{Z}_s$ is built as previously.

**Temporal bounds.** Given temporal boundaries for each action instance, we first add all tracklets that are outside of these ground truth intervals to $\mathcal{O}_s$ in order to assign them to the background class. Then, we augment $\mathcal{B}_s$ with one bag per time unit composing the ground truth interval in order to ensure that at least one tracklet is assigned to the corresponding action class at all time in the ground truth interval. The set $\mathcal{Z}_s$ is constructed as above.

**Temporal bounds with bounding boxes.** In addition to temporal boundaries of action instances, we are also given a few frames with spatial annotation. This is a standard scenario adopted in DALY [46] and AVA [13]. In our experiments, we report results with one and three bounding boxes per action instance. $\mathcal{Z}_s$ and $\mathcal{B}_s$ are constructed as in the '**Temporal bounds only**' setting. $\mathcal{O}_s$ is initialized to assign all tracklets outside of the temporal intervals to the background class. Similarly to the '**One bounding box**' scenario, we augment $\mathcal{O}_s$ in order to force tracklets to belong to either the corresponding action or the background depending on the spatial overlap criterion. However, here an action instance has potentially several annotations, therefore the spatial overlap with the track is defined as the minimum IoU between the corresponding bounding boxes.

**Fully supervised.** Here, we enforce a hard assignment of all tracklets to action classes based on the ground truth through $\mathcal{O}_s$. When a tracklet has a spatial IoU greater than 0.3 with at least one ground truth instance, we assign it to the action instance with the highest IoU. Otherwise, the tracklet is assigned to the background.

**Temporal bounds with spatial points.** In [27], the authors introduce the idea of working with spatial points instead of bounding boxes for spatial supervision. Following [27], we take the center of each annotated bounding box to simulate the spatial point annotation. Similarly to the '**Fully supervised**' setting, we build $\mathcal{O}_s$ to enforce a hard assignment for all tracklets, but we modify the action-tracklet matching criterion. When tracklet bounding boxes contain all the corresponding annotation points of at least one ground truth instance, we assign the tracklet to the action instance that has the lowest distance between the annotation point and the bounding box center of the tracklet.

### 4.4 Results and analysis

In this section we present and discuss our experimental results. We evaluate our method on two datasets, UCF101-24 and DALY, for the following supervision levels: video level, temporal point, one bounding box, temporal bounds only, temporal bounds and one bounding box, temporal bounds and three bounding boxes. For UCF101-24, we also report temporal bounds and spatial points [27] and the fully supervised settings (not possible for DALY as the spatial annotation is not dense). We evaluate the additional 'shot-level' supervision setup on DALY. All results are given in Table 1. We compare to the state of the art whenever possible. To the best of our knowledge, previously reported results are not available for all levels of supervision considered in this paper. We also report results for the mixture of different levels and amounts of supervision in Figure 2a. Qualitative results are illustrated in Figure 2b and in the Appendix.

**Comparison to the state of the art.** We compare results to two state-of-the-art methods [27, 46] that are designed to deal with weak supervision for action localization. Mettes et al. [27] use spatial point annotation instead of bounding boxes. Weinzaepfel et al. [46] compare various levels of spatial supervision (spatial points, few bounding boxes). Temporal boundaries are known in both cases. We also compare our fully supervised approach to recent methods, see Table 2.

**UCF101-24 specificity.** For UCF101-24, we noticed a significant drop of performance whenever we use an off-the-shelf detector versus a detector pretrained on UCF101-24 bounding boxes. We observe that this is due to two main issues: (i) the quality of images in the UCF101-24 dataset is quite low when compared to DALY which makes the human detection very challenging, and (ii) the bounding boxes have been annotated with a large margin around the person whereas a typical off-the-shelf detector produces tight detections (see Appendix). Addressing the problem (i) is difficult without adaptive finetuning. Concerning (ii), a simple solution adopted in our work is to enlarge person

| Supervision | | Video level | Shot level | Temporal point | Temporal | Temporal + spatial points | | | 1 BB | Temp. + 1 BB | | Temp + 3 BBs | | Fully Supervised | |
|---|---|---|---|---|---|---|---|---|---|---|---|---|---|---|---|
| Method | | Our | Our | Our | Our | Our | [46] | [27] | Our | Our | [46] | Our | [46] | Our | [46] |
| UCF101-24 | @0.2 | 43.9 | - | 45.5 | 47.3 (69.5) | 49.1 (69.8) | 57.5 | 34.8 | 66.8 | 70.6 | 57.4 | 74.5 | 57.3 | 76.0 | 58.9 |
| (mAP) | @0.5 | 17.7 | - | 18.7 | 20.1 (38.0) | 19.5 (39.5) | - | - | 36.9 | 38.6 | - | 43.2 | - | 50.1 | - |
| DALY | @0.2 | 7.6 | 12.3 | 26.7 | 31.5 (33.4) | No continuous spatial GT | | | 28.1 | 32.5 | 14.5 | 32.5 | 13.9 | No full GT available | |
| (mAP) | @0.5 | 2.3 | 3.9 | 8.1 | 9.8 (14.3) | | | | 12.2 | 13.3 | - | 15.0 | - | | |

Table 1: Video mAP for varying levels of supervision.

detections with a single scaling factor (we use $\sqrt{2}$). However, we also observe that the size of boxes depends on action classes (e.g. the bounding boxes for the 'TennisSwing' contains the tennis racket), which is something we cannot capture without more specific information. To better highlight this problem we have included in Table 1 a baseline where we use the detections obtained after finetuning on spatial annotations even when it is normally not possible ('temporal' and 'temporal with spatial points' annotations). The detector is the same as in the 'one bounding box' supervision setup. We report these baselines in parenthesis in Table 1. We can observe that the drop of performance is much higher on UCF101-24 ($-22.2\%$ mAP@0.2 for 'temporal') than on DALY ($-1.9\%$), which confirms the hypothesis that the problem actually comes from the tracks rather than from our method.

**Results for varying levels of supervision.** Results are reported in Table 1. As expected, the performance increases when more supervision is available. On the DALY dataset our method outperforms [46] by a significant margin ($+18\%$). This highlights the strength of our model in addition to its flexibility. We first note that there is only a $1.0\%$ difference between the 'temporal' and 'temp + 3 keys' levels of supervision. This shows that, with recent accurate human detectors, the spatial supervision is less important. Interestingly, we observe good performance when using one temporal point only ($26.7\%$). This weak information is almost sufficient to match results of the precise 'temporal' supervision ($31.5\%$). The corresponding performance drop is even smaller on the UCF101-24 dataset ($-1.8\%$). This demonstrates the strength of the single 'temporal click' annotation which could be a good and cheap alternative to the precise annotation of temporal boundaries.

On the UCF101-24 dataset, we are always better compared to the state of the art when bounding box annotations are available, e.g., $+17.2\%$ in 'temp + 3 keys' setting. This demonstrates the strength of the sparse bounding box supervision which almost matches accuracy of the fully-supervised setup ($-1.5\%$). We note that using only one bounding box already enables good performance ($66.8\%$). Overall, this opens up the possibility to annotate action video datasets much more efficiently at a moderate cost. However, as explained above, we observe a significant drop when removing the spatial supervision. This is mainly due to the fact that [46] are using more robust tracks obtained by a sophisticated combination of detections and an instance specific tracker. This is shown in Table 1 of the Appendix where we run our approach with tracks of [46]. We obtain better results than [46] in all cases and a video level mAP of $53.1\%$. 'Video level' is reported in [26]. Compared to their approach, which is specifically designed for this supervision, we achieve better performance on UCF101-24 ($43.9\%$ ($53.1\%$ with tracks from [46]) vs. $37.4\%$).

**Comparison to fully supervised baselines.** Table 2 reports additional baselines in the fully supervised setting for the UCF101-24 dataset. Note that, even if not the main focus of our work, our performance in the fully-supervised settings is on par with many recent approaches, e.g. the recent work [18] which reports a mAP@0.5 of $49.2\%$ on UCF101-24 (versus our $50.1\%$). This shows again that our model is not only designed for weak supervision, and thus can be used to fairly compare all levels of supervision. However, we are still below the current state of the art [13] on UCF101-24 with full supervision, which reports a mAP@0.5 of $59.9\%$ on UCF101-24 (versus our $50.1\%$). This can be explained by the fact that we are only learning a linear model for I3D features, whereas [13] train a non-linear classifier for the same features.

| Video mAP | @0.2 | @0.5 |
|---|---|---|
| [46] | 58.9 | - |
| Peng w/ MR [31] | - | 35.9 |
| Singh *et al.* [37] | 73.5 | 46.3 |
| ACT [18] | 76.5 | 49.2 |
| AVA [13] | - | 59.9 |
| Our method | 76.0 | 50.1 |

Table 2: Comparison of fully supervised methods on UCF101-24.

Investigating how to build flexible deep (i.e. non-linear) models that can handle varying level of supervision is therefore an interesting avenue for future work.

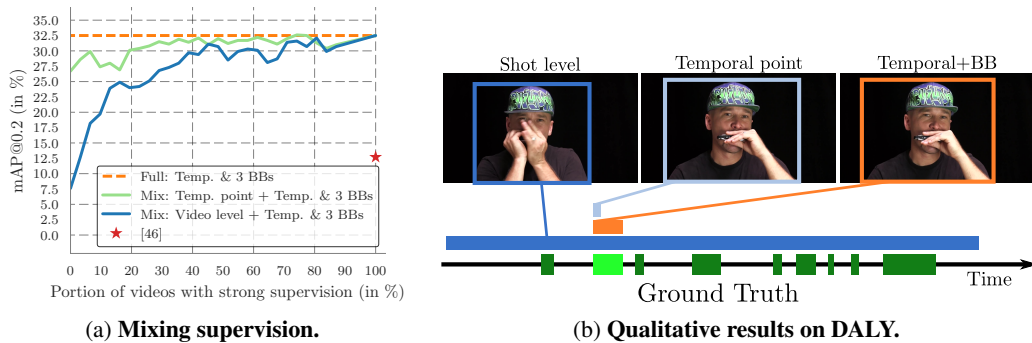

| (a) **Mixing supervision.** | (b) **Qualitative results on DALY.** |

Figure 2: **Left.** Mixing levels of supervision on the DALY dataset. **Right.** Predictions for the light green ground truth instance for various levels of supervision. We display the highest scoring detection in each case. When training with only 'shot-level' annotation it is hard for the method to discover precise boundaries of actions, as 'Playing Harmonica' could also consist in someone holding an harmonica. Annotating with a temporal point is sufficient to better detect the instance. Training with precise temporal boundaries further improves performance.

**Mixing levels of supervision.**    To further demonstrate the flexibility of our approach, we conduct an experiment where we mix different levels of supervision in order to improve the results. We consider the weakest form of supervision, i.e. the 'video-level' annotation and report results on the DALY dataset. The experimental setup consists in constructing a training dataset with a portion of videos that have weak supervision (either 'video-level' or 'temporal point') and another set with stronger supervision (temporal bounds and 3 bounding boxes). We vary the portions of videos with stronger supervision available at the training time (the rest having weak labeling), and we evaluate mAP@0.2 on the test set. We report results in Figure 2a. Tracks are obtained using the off-the-shelf person detector. With only 20 supervised videos (around $5\%$ of the training data) and 'video level' labels for remaining videos, the performance goes from 7.6 to 18.2. We are on par with the performance in the fully supervised setting when using only $40\%$ of the fully annotated data. This performance is reached even sooner when using 'Temporal point' weak labeling (with only $20\%$ of fully annotated videos). This strongly encourages the use of methods with the mixed levels of supervision for action localization.

## 5    Conclusion

This paper presents a weakly-supervised method for spatio-temporal action localization which aims to reduce the annotation cost associated with fully-supervised learning. We propose a unifying framework that can handle and combine varying types of less-demanding weak supervision. The key observations are that **(i)** dense spatial annotation is not always needed due to the recent advances of human detection and tracking, **(ii)** the performance of 'temporal point' supervision indicates that only annotating an action with a 'click' is very promising to decrease the annotation cost at a moderate performance drop, and **(iii)** mixing levels of supervision (see Figure 2a) is a powerful approach for reducing annotation efforts.

## Acknowledgements

We thank Rémi Leblond for his thoughtful comments and help with the paper. This work was supported in part by ERC grants ACTIVIA and ALLEGRO, the MSR-Inria joint lab, the Louis Vuitton ENS Chair on Artificial Intelligence, an Amazon academic research award, the Intel gift and the DGA project DRAAF.

## Footnotes

[1]Inria, École normale supérieure, CNRS, PSL Research University, 75005 Paris, France.

[2]University Grenoble Alpes, Inria, CNRS, Grenoble INP, LJK, 38000 Grenoble, France.

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
