[Supplementary Material · appendix.pdf]

# Appendix: A flexible model for training action localization with varying levels of supervision

## 1 UCF101-24 specificity

Figure 1 illustrates the UCF101-24 specificity described in Section 4.4 of the main paper.

Figure 1: UCF101-24 specificity. A typical person detector (cyan) results in tight bounding boxes around the person, while the UCF101-24 ground-truth action annotations (green) often have large spatial extents and may include objects such as the tennis racket. This reduces significantly the IoU measure between predictions and ground truth (see numbers in parenthesis). Class-specific annotation biases can be learned e.g. by a fine-tuned detector (magenta) as explained in Section 4.4 of the main paper. The detection scaling procedure (blue) works in some cases but this simple linear expansion is not enough to reach satisfying precision. The IoU score of detectors w.r.t. the ground truth is indicated in brackets.

To further illustrate this point we have also conducted an additional experiment. We have rerun our method with the same tracks as used in [1] ([46] in the paper) for all types of supervision on UCF101-24. These results are presented in Table 1 (our w. tracks of [1]) and compared to [1]. We observe that "Our w. tracks of [1]" outperforms [1] in *all settings*, and particularly for the "Temporal+spatial points" setup. As discussed in the paper, the tracks of [1] are better adapted to handle the specificity of the UCF101-24 annotation. This observation is well confirmed in Table 1 where "Our w. tracks from [1]" outperforms our method using a generic person detector in "Ours" (col. 2–5). When the bounding box supervision is available, however, the adjustment of our generic tracks for specific actions improves results beyond "Our w. tracks from [1]" (col. 6–9). Finally, when using the same tracklets for all settings (Our w. tracks of [46]), we observe only a small difference in performance between the "Temporal" and the "Temporal + 1 BB" setups. This suggests an interesting conclusion

that the spatial supervision for action localization may not always be necessary as highlighted in the conclusion of the paper.

| Supervision | Video level | Temporal point | Temporal | Temporal + spatial points | 1 BB | Temp. + 1 BB | Temp. + 3 BBs | Fully Supervised |
|---|---|---|---|---|---|---|---|---|
| [1] | - | - | - | 57.5 | - | 57.4 | 57.3 | 58.9 |
| Our w. tracks of [1] | **53.1** | **54.7** | **59.7** | **61.0** | 61.2 | 58.9 | 62.1 | 61.2 |
| Our (see paper) | 43.9 | 45.5 | 47.3 (69.5) | 49.1 (69.8) | **66.8** | **70.6** | **74.5** | **76.0** |

Table 1: Video mAP@0.2 on UCF101-24 for varying levels of supervision with different set of tracks.

## 2 Qualitative results

Some qualitative results are presented for DALY (Figure 2 and Figure 3) and for UCF101-24 (Figure 4 and Figure 5) datasets. For each video, we choose an action instance (shown in light green) and display action predictions that have been matched to this instance for three different models trained with varying levels of supervision. Other ground truth instances are displayed in dark green. For DALY, we report predictions made by models trained from 'Shot level' (dark blue), 'Temporal point' (light blue) and 'Temporal and bounding box (BB)' (orange) supervisions. For UCF101-24, we report 'Video level' (dark blue), 'Temporal and bounding box (BB)' (light blue) and 'Fully supervised' (orange). The image and the box displayed for each method are the one that obtained the highest score of detection according to each model. On UCF101-24, the ground-truth bounding box is displayed in green.

## References

[1] Philippe Weinzaepfel, Xavier Martin, and Cordelia Schmid. Human action localization with sparse spatial supervision. In *CoRR*, 2016. 1, 2

Figure 2: Qualitative results for spatio-temporal localization on DALY. The timeline indicates GT (green), shot level supervision results (dark blue), temporal point supervision results (light blue) and temporal + 1 BB supervision results (orange).

Figure 3: Qualitative results for spatio-temporal localization on DALY. The timeline indicates GT (green), shot level supervision results (dark blue) and temporal point supervision results (light blue) and temporal + 1 BB supervision results (orange).

Figure 4: Qualitative results for spatio-temporal localization on UCF101-24. The timeline indicates GT (green), video level supervision results (dark blue) and temporal + 1 BB supervision results (light blue) and full supervision results (orange).

Figure 5: Qualitative results for spatio-temporal localization on UCF101-24. The timeline indicates GT (green), video level supervision results (dark blue) and temporal + 1 BB supervision results (light blue) and full supervision results (orange).