[Reviews · NeurIPS 2018]

Reviewer 1



The authors introduce a spatiotemporal action detection approach using weak supervision. Namely, an L2 norm-based discriminative clustering approach is coupled with different types of supervision as constraints. In particular, such constraints are related to video-level class labels, temporal points or bounding boxes. In turn, a regularized cost function under a linear model is solved based on the Frank-Wolfe algorithm to relax integer constraints. Attained results show an interesting flexibility regarding the weak supervision, which is useful for many computer vision applications. The paper is well-motivated, however, the mathematical derivations seem to be fair for a NIPS paper. In particular, a deep analysis of Eq (5) could be included concerning the optimization foundations. Besides, the implementation and mathematical details of the supervision as constraints section could be enhanced (at least in the supplementary material).

Reviewer 2



Paper Summary: The paper describes a method for spatio-temporal human action localization in temporally untrimmed videos based on discriminative clustering [3, 47]. The main contribution of this paper is a new action detection approach which is flexible in the sense that it can be trained with various levels and amounts of supervision. For example, the model can be trained with very weak level of supervision, i.e., train the model for action detection only using ground truth video-level action labels; and also it can be trained with full supervision i.e. with dense per frame bounding box and their class labels. Experimental results demonstrate the strengths and weaknesses for a wide range of supervisory signals such as, video level action labels, single temporal point, one GT bounding box, temporal bounds etc. The method is experimentally evaluated on the UCF-101-24 and DALY action detection datasets. Paper Strengths: - The paper is clear and easy to understand. - The problem formulation is interesting and described with enough details. - The experimental results are interesting and promising, clearly demonstrate the significance of varying level of supervision on the detection performance - Table 1. - On DALY dataset, as expected, the detection performance increases with access to more supervision. - The proposed approach outperforms the SOA [46] by a large margin of 18% (video mAP) on DALY dataset at all levels of supervision. - On UCF-101-24, the proposed approach outperforms the SOA [46] when bounding box annotations are available at any level, i.e., Temp.+1 BB, Temp. + 3 BBs, Fully Spervised (cf. Table 1). - The visuals are helpful, support well the paper, and the qualitative experiments (in supplementary material) are interesting and convincing. Paper Weaknesses: I haven't noticed any major weakness in this paper, however would like to mention that - on UCF-101-24, the proposed method has drop in performance as compared to the SOA [46] when supervision level is "Temporal + spatial points". This work addresses one of the major problems associated with action detection approaches based on fully supervised learning, i.e., these methods require dense frame level GT bounding box annotations and their labels, which is impractical for large scale video datasets and also highly expensive. The proposed unified action detection framework provides a way to train a ML model with weak supervision at various levels, contributing significantly to address the aforementioned problem. Thus, I vote for a clear accept.

Reviewer 3



This paper present a framework to use different level of weakly supervised signals for training action localization models. The problem of assigning tracklets to labels is formulated as a discriminative clustering problem with constraints on the tracklet labels generated from weakly supervised signals. This paper presented experimental results showing the effectiveness of different weak supervision signals and comparison with state-of-the-art. Strength: well written with algorithm details and implementation details, clear experiment section and contribution to the community: 1. experimental results help to understand trade-offs of different type of annotations for videos 2. a new weakly supervised algorithm for video localization. weakness/questions: 1. Description of the framework: It's not very clear what Bs is in the formulation. It's not introduced in the formulation, but later on the paper talks about how to form Bs along with Os and Zs for different supervision signals. And it;s very confusing what is Bs's role in the formulation. 2. computational cost: it would be great to see an analysis about the computation cost. 3. Experiment section: it seems that for the comparison with other methods, the tracklets are also generated using different process. So it's hard to draw conclusions based on the results. Is it possible to apply different algorithms to same set of tracklets? For example, for the comparison of temporal vs temp+BB, the conclusion is not clear as there are three ways of generating tracklets. It seems that the conclusion is -- when using same tracklet set, the temp + BB achieves similar performance as using temporal signal only. However, this is not explicitly stated in the paper. 4. The observation and conclusions are hidden in the experimental section. It would be great if the paper can highlight those observations and conclusions, which is very useful for understanding the trade-offs of annotation effort and corresponding training performance. 5. comparison with fully supervised methods: It would be great if the paper can show comparison with other fully supervised methods. 6. What is the metric used for the video level supervision experiment? It seems it's not using the tracklet based metrics here, but the paper didn't give details on that.